# Uncertainty of Interval Type-2 Fuzzy Sets Based on Fuzzy Belief Entropy

**DOI:** 10.3390/e23101265

**Published:** 2021-09-28

**Authors:** Sicong Liu, Rui Cai

**Affiliations:** 1College of Computer and Information Science, Southwest University, Chongqing 400700, China; liusicongswu@163.com; 2College of Business and Commerce, Rongchang Campus, Southwest University, Chongqing 402460, China

**Keywords:** interval type-2 fuzzy sets, uncertainty measure, fuzzy belief entropy, D–S evidence theory, Z-valuations

## Abstract

Interval type-2 fuzzy sets (IT2 FS) play an important part in dealing with uncertain applications. However, how to measure the uncertainty of IT2 FS is still an open issue. The specific objective of this study is to present a new entropy named fuzzy belief entropy to solve the problem based on the relation among IT2 FS, belief structure, and Z-valuations. The interval of membership function can be transformed to interval BPA [Bel,Pl]. Then, Bel and Pl are put into the proposed entropy to calculate the uncertainty from the three aspects of fuzziness, discord, and nonspecificity, respectively, which makes the result more reasonable. Compared with other methods, fuzzy belief entropy is more reasonable because it can measure the uncertainty caused by multielement fuzzy subsets. Furthermore, when the membership function belongs to type-1 fuzzy sets, fuzzy belief entropy degenerates to Shannon entropy. Compared with other methods, several numerical examples are demonstrated that the proposed entropy is feasible and persuasive.

## 1. Introduction

Nowadays, the study of uncertainty has received extensive attention [1,2]. Discord, nonspecificity, and fuzziness are three main types of uncertainty [3]. In addition, discord and nonspecificity are classified into the ambiguity (Figure 1). In order to handle various types of uncertainty, many theories are developed, such as fuzzy sets [4], D–S evidence theory [5,6], rough sets [7], interval evidence theory [8,9,10,11], Z numbers [12], soft sets [13], and D numbers [14].

Zadeh proposed type-1 fuzzy sets (T1 FS) [4] in 1965, and T1 FS can better model a single user’s understanding of semantic concepts [15], and it can effectively solve practical problems [16,17]. However, different people may have different understandings of the same semantic concept [18]. For the purpose of handling the uncertainty between individuals efficiently, some scholars put forward many forms of fuzzy sets such as type-2 fuzzy sets (T2 FS) [19] and intuitionistic fuzzy sets (IFS) [20,21]. Because the expression and understanding of T2 FS are complicated, the most commonly used is the special form of T2 FS, namely interval type-2 fuzzy sets (IT2 FS) [22,23]. For each element in IT2 FS, the membership becomes an interval rather than a definite value. Due to its efficiency to handle uncertainty, the application of IT2 FS is more and more widespread such as decision-making [24,25,26], time-series forecasting [27], speech recognition [28], and others [29,30]. In addition, fuzzy entropy has been widely used in practical applications [31,32,33]. Some studies on the uncertainty measure of IT2 FS are given attention: Initially, some definitions of the fuzziness for IT2 FS are presented [34,35] and most of the existing works are based on IFS and T1 FS [21]. However, fuzziness is only a part of the measurement of uncertainty [3], and there are some other important properties of IT2 FS, such as centroid, cardinality, variance, and skewness. In the recent uncertain studies on IT2 FS, Mendel’s method [36] and Green’s method [37] have attracted wide attention. In Mendel’s research, uncertainty measures are given for five important characteristics of IT2 FS. The calculated result is the interval value of five indicators, not a total measurement value, and the length of the interval is used as the measurement uncertainty [36]. Greenfield [37] argues that the area of the footprint can be used to measure the uncertainty of IT2 FS. In addition, the centroid length measured by Mendel’s method and the area measured by Greenfield’s method are compared, which shows the similar results [37].

The imprecise representation in the Dempster–Shafer model has the same idea as the interval membership in the type-2 fuzzy sets [38,39,40]. For instance, both D–S evidence theory and IT2 FS use intervals to indicate the inaccuracy of information. Furthermore, the relation among T2 FS, Z-valuations, and evidence theory is also found [41], so that IT2 FS can be expressed by belief structure. As a method for handling uncertain problems, D–S evidence theory has received widespread attention [42]. Basic probability assignment (BPA) is the key parameter for belief structure. Uncertainty in evidence theory can be measured by two indicators [43,44]: non-specificity and discord, which can be effectively expressed by interval BPA. Entropy is the preferred mathematical quantity to express uncertainty [45], and various types of entropy have been proposed, such as Shannon entropy [46], Tsallis entropy [47], fuzzy entropy [48,49], and nonadditive entropy [50]. Recently, a new method for measuring uncertainty degree of BPA, Deng entropy [51], was proposed. Different from other methods, Deng entropy considers both non-specificity and discord [52], which is more persuasive. When BPA is degenerated as probability distribution, Deng entropy degenerates to Shannon entropy [51]. However, Deng entropy cannot measure the uncertainty for an interval of belief structure. There have been some studies on interval belief entropy [52,53]. Moreover, the interval size could be considered as the fuzziness measure in the uncertainty measurement of interval belief structure [53].

Mendel’s method [36] and Greenfield’s method [37] have a certain rationality in measuring the uncertainty of IT2 FS. Nonetheless, there are still obvious drawbacks to the two methods. First of all, all of the above methods did not consider the uncertainty caused by the multiple subsets, if the membership functions of a single subset and a multiple subset are the same, the uncertainty calculated by the two methods is also the same, which is not reasonable because multiple subsets contain more uncertainty and the value should be bigger. In addition, Mendel’s method needs to obtain the upper and lower bounds of the measured values of all points through the search algorithm [37], which requires extra calculation to obtain the required information. In engineering, this means that the calculation is very large, resulting in low efficiency.

To address the issue, we consider from the perspective of belief structure and propose a new entropy named fuzzy belief entropy to measure uncertainty, the main objective of which is not only to avoid the large amount of computation, but also to measure the uncertainty caused by multivariate subsets. Our research process is shown as follows: first, derived from the relation among T2 FS, D–S evidence theory, and Z-valuations, the membership function of IT2 FS could be converted into interval BPA. Then, we present a new entropy named fuzzy belief entropy to handle the uncertainty. Fuzzy belief entropy measures the uncertainty from the three aspects of fuzziness, discord, and nonspecificity, respectively. In addition, when the membership function belongs to type-1 fuzzy sets, fuzzy belief entropy degenerates to Shannon entropy.

The paper is organized as follows: in Section 2, IT2 FS, D–S evidence theory, the relation among IT2 FS, D–S evidence theory, and some belief entropy are discussed. In Section 3, fuzzy belief entropy is proposed and some characteristics are listed. In Section 4, different types of examples are given and compared with other methods to demonstrate the feasibility of fuzzy belief entropy. In Section 5, the works of this article are summed up.

## 2. Preliminaries

In the section, the essential concepts including IT2 FS, D–S evidence theory, and the relationship between IT2 FS and belief structure are briefly introduced.

### 2.1. Interval Type-2 Fuzzy Sets

**Definition** **1.**
*Type-2 fuzzy sets (T2 FS), denoted $\tilde{A}$, which is defined as [19]:*

(1)
$$\tilde{A}\;\;=\left\{\left(\left(x,u\right),\mu_{\tilde{A}}\left(x,u\right)\right)\left|\;\;x\in X\right.,\forall u\in J_{x}\subseteq \left[0,1\right]\right\}$$

*where $\mu_{\tilde{A}}\left(x,u\right)$ is a type-2 membership function and it satisfies that $0\leq \mu_{\tilde{A}}\left(x,u\right)\leq 1$. In addition, there’s another form of $\tilde{A}$, and can be represented as:*

(2)
$$\tilde{A}\;\;=\;\;\int_{x\in X}\int_{u\in J_{x}}\mu_{\tilde{A}}\left(x,u\right)/\left(x,u\right)\;\;J_{x}\subseteq \left[0,1\right]$$



**Definition** **2.**
*Interval type-2 fuzzy set (IT2 FS), which satisfies $\mu_{\tilde{A}}\left(x,u\right)=1$, is the special form of T2 FS. $\tilde{A}$ is defined as [22]:*

(3)
$$\tilde{A}\;\;=\;\;\int_{x\in X}\int_{u\in J_{x}}1/\left(x,u\right)\;\;,J_{x}\subseteq \left[0,1\right]$$



**Definition** **3.**
*The membership function of $\tilde{A}$ is three-dimensional. When $\tilde{A}$ is mapped to a two-dimensional plane, the area covered by the interval boundary of membership function is referred to as the footprint of uncertain (FOU) of $\tilde{A}$ (Figure 2). FOU is defined as [23]:*

(4)
$$FOU\left(\tilde{A}\right)\;\;=\;\;\underset{x\in X}{⋃}J_{x}$$



### 2.2. Basic Concepts in Belief Structure

As a classical data fusion theory, D–S evidence theory is widely applied in the field of uncertainty [54,55]. Here are some essential concepts:

#### 2.2.1. Frame of Discernment (FOD)

In evidence theory, a frame of discernment Θ is a sample space, which is composed of two mutually exclusive elements and contains all objects, and is defined as [5]:(5)$$\Theta =\left\{\theta_{1},\theta_{2},\cdots ,\theta_{i},\cdots ,\theta_{N}\right\}$$

There are *N* elements in FOD, and the power set $2^{\Theta }$ which includes $2^{N}$ elements is defined as:(6)$$2^{\Theta }=\left\{⊘,\left\{\theta_{1}\right\},\left\{\theta_{2}\right\},\cdots ,\left\{\theta_{1},\theta_{2}\right\},\cdots ,\Theta \right\}$$

#### 2.2.2. Basic Probability Assignment (BPA)

BPA is defined as a mapping relation *m*:$2^{\Theta }$ → [0, 1] and satisfied [5,6]:(7)$$m\left(⊘\right)=0\;\;\sum_{A\in 2^{\Theta }}m\left(A\right)=1$$
where proposition $A\in 2^{\Theta }$. If $m\left(A\right)>0$ is satisfied, then *A* is a focal element.

#### 2.2.3. Belief Function

Suppose *A* is a proposition in the power set $2^{\Theta }$. The belief function Bel(A) means the full point of confidence given to proposition *A*. Bel(A) is defined as [5,6]:(8)$$Bel\left(A\right)=\sum_{B\subseteq A}m\left(B\right)\;\;\forall A\subseteq \Theta$$

#### 2.2.4. Plausibility Function

Suppose *A* is a proposition in the power set $2^{\Theta }$. The plausibility function Pl(A) indicates the degree that it does not object to that proposition *A* is true. Pl(A) is defined as [5,6]:(9)$$Pl\left(A\right)=\sum_{B\cap A\neq ⊘}m\left(B\right)=1-\sum_{B\cap A=⊘}m\left(B\right)\;\;\forall A\subseteq \Theta$$

### 2.3. The Relation among IT2 FS, Z-Valuations, and Evidence Theory

Suppose there is a variable *v* with domain *X*, which is indicated by belief structure *m*. In addition, *A* be any fuzzy subset of *X*. Assume that the probability of *A* is unknown, but the interval of the probability can be obtained. Then, the lower and upper bounds on the probability of *A* are named as the belief and plausibility of *A* by Shafer. We can express their relation in the following form [41]:

(1) The belief function is
(10)$$Prob_{m}^{-}\left(A\right)=Bel_{m}\left(A\right)=\sum_{j=1}^{N}\left(1-Max_{x}\left[D_{j}\left(x\right)\bar{A}\left(x\right)\right]\right)\alpha_{j}$$

(2) The plausibility function is
(11)$$Prob_{m}^{+}\left(A\right)=Pl_{m}\left(A\right)=\sum_{j=1}^{N}\left(Max_{x}\left[A\left(x\right)D_{j}\left(x\right)\right]\right)\alpha_{j}$$
where $D_{j}$ is the subset from j=1 to j=N; $\alpha_{j}$ and $m\left(D_{j}\right)$ are the unspecified amounts of probability, which are shared among the elements in $D_{j}$ [38].

The belief and plausibility are adopted to indicate the imprecise probability of *A*. Therefore, the probability belongs to an interval denoted by $Prob_{m}\left(A\right)=\left[Bel\left(A\right),Pl\left(A\right)\right]$, which satisfies $Prob_{m}\left(A\right)\subseteq \left[0,1\right]$ [38].

The above mentioned is the connection between T1 FS and belief structure. Furthermore, on the basis of the above representation, Z-valuation is considered to get the result we need. There is a Z-valuation, zis(A,B), which means the probability of (z=A) is *B*. Assume there is a variable *v*, which can be represented by a Z-valuation [12] zis(A,B). Under belief structure *m*, the probability of *A* becomes an interval value: $Prob_{m}\left(A\right)=\left[Bel_{m}\left(A\right),Pl_{m}\left(A\right)\right]$. Furthermore, we intend to obtain the range for $B\left(Prob_{m}\left(z\;\;=\;\;A\right)\right)$ [38]:(12)$$B\left(Prob_{m}\left(z\;\;=\;\;A\right)\right)=\left[L_{m},U_{m}\right]$$
where *B* is the confidence value; $L_{m}={\mathrm{Min}}_{r\in R_{m}\left(A\right)}\left[B\left(r\right)\right]$ and $U_{m}=Max_{r\in R_{m}\left(A\right)}\left[B\left(r\right)\right]$. In this case, for any m∈X, we have $F\left(m\right)=\left[L_{m},U_{m}\right]$. In particular, *F* could be expressed derived from the definition of T2 FS [22,41]:(13)$$F=\left\{\left(z,p\left(A\right)\right),B\left(p\left(A\right)\right)\left|\;\;\forall A\in X,p\left(A\right)\subseteq \left[0,1\right],B\left(p\left(A\right)\right)\in \left[0,1\right]\right.\right\}$$
where $B\left(p\left(A\right)\right)$ is the type-2 membership function of *F*.

Based on this, the meaning of the Z-valuation expression can be understood as the probability distribution of fuzzy subset *F* under belief structure *m*. When *B* is a positive confidence value, $B\left(x\right)\geq B\left(y\right)$ for x>y. The lower and higher bounds on the probability are $L_{m}=B\left(Bel_{m}\left(A\right)\right)\;\;$ and $U_{m}=\;\;B\left(Pl_{m}\left(A\right)\right)$ [41]. Moreover, when *F* is an IT2 FS, $B\left(p\left(A\right)\right)=1$, which shows the information Prob (*z* = *A*) is 1 in the Z-valuations. Therefore, we can get the result that $F\left(m\right)=\left[L_{m},U_{m}\right]=\left[Bel_{m}\left(A\right),Pl_{m}\left(A\right)\right]$, which represents that the two intervals are the same. Thus, T2 FS can be expressed by belief structure.

### 2.4. Shannon Entropy

Shannon entropy is the basis of information theory, which can effectively describe the uncertainty and information capacity of a probability distribution. It mainly measures the chaotic degree of the system and represents the sum of information in the system. If the entropy is bigger, it means that the uncertainty is bigger, so we need to get more information. Then, Shannon entropy is defined as [46]:(14)$$H_{S}=-\sum_{i=1}^{N}p_{i}\log_{b}p_{i}$$

### 2.5. Existing Belief Entropy

For belief structure, the discord and the nonspecificity are two types of uncertainty simultaneously [43]. As a measure of discord, Shannon entropy is generally applied in describing the uncertainty of probability distribution. A lot of scholars proposed new measure functions based on Shannon entropy. Typical examples are shown in Table 1.

Dubois and Prade’s UM is a method for measuring non-specificity for belief structure. Other methods are the measure of discord [51]. In addition, Deng entropy considered uncertainty in terms of both Discord and the nonspecificity, but it cannot directly measure the uncertainty for a given belief interval. Wang & Song’s UM and Pan & Deng’s entropy both use the intermediate value of the belief interval to measure the uncertainty, and the former holds that the length of interval can be used as a measure of fuzziness, while the latter is improved on the basis of Deng entropy, so the uncertainty brought by multivariate subsets is considered. However, these two aspects are very important to measure the uncertainty of IT2 FS. In this paper, a new entropy is proposed, which measures the uncertainty caused by both interval length and multivariate subset.

## 3. Fuzzy Belief Entropy

As mentioned in Section 2.3, we suppose that *F* is an IT2 FS over a frame of discernment Θ. Then, it can be expressed as: $F\left(m\right)=\left[L_{m},U_{m}\right]=\left[Bel_{m}\left(A\right),Pl_{m}\left(A\right)\right]$. Therefore, the interval of fuzzy membership function could be converted to belief structure, with the lower and upper bounds as the belief function Bel and the plausibility function Pl. Then, we can get the uncertainty of IT2 FS by calculating the uncertainty of interval BPA. In this study, combining the relationship between IT2 FS and interval BPA, a new entropy called fuzzy belief entropy is proposed to measure uncertainty, which is applicable to both IT2 FS and belief structure. Fuzzy belief entropy is defined as:(15)$$U_{F}\left(m\right)=\sum_{\theta_{i}\in 2^{\Theta }}\left[-\frac{Bel\left(\theta_{i}\right)+Pl\left(\theta_{i}\right)}{2}\log_{2}\frac{Bel\left(\theta_{i}\right)+Pl\left(\theta_{i}\right)}{2\left(2^{\left|\theta_{i}\right|}-1\right)}+\frac{Pl\left(\theta_{i}\right)-Bel\left(\theta_{i}\right)}{2}\right]$$
where *m* denotes a mass function and the proposition $\theta_{i}$ is a fuzzy subset of Θ; $\left|\theta_{i}\right|$ is the cardinal number of $\theta_{i}$; $Bel\left(\theta_{i}\right)$ and $Pl\left(\theta_{i}\right)$ are the belief function and plausibility function of $\theta_{i}$. Through an equivalent transformation, we can better observe some properties of fuzzy belief entropy, as follows:(16)$$\begin{array}{ccc} U_{F}\left(m\right) & = & \sum_{\theta_{i}\in 2^{\Theta }}\begin{array}{c} \frac{Bel\left(\theta_{i}\right)+Pl\left(\theta_{i}\right)}{2}\log_{2}\left(2^{\left|\theta_{i}\right|}-1\right)\;\; \end{array}\\ & & \;+\sum_{\theta_{i}\in 2^{\Theta }}\frac{Bel\left(\theta_{i}\right)-Pl\left(\theta_{i}\right)}{2}\\ & & \;-\sum_{\theta_{i}\in 2^{\Theta }}\frac{Bel\left(\theta_{i}\right)+Pl\left(\theta_{i}\right)}{2}\log_{2}\frac{Bel\left(\theta_{i}\right)+Pl\left(\theta_{i}\right)}{2} \end{array}$$
where the term $\sum_{\theta_{i}\in 2^{\Theta }}\begin{array}{c} \frac{Bel\left(\theta_{i}\right)+Pl\left(\theta_{i}\right)}{2}\log_{2}\left(2^{\left|\theta_{i}\right|}-1\right) \end{array}$ represents the part of measuring nonspecificity in the mass function *m*.

$\sum_{\theta_{i}\in 2^{\Theta }}\frac{Bel\left(\theta_{i}\right)-Pl\left(\theta_{i}\right)}{2}$ represents the part of measuring fuzziness in the mass function *m*.

$-\sum_{\theta_{i}\in 2^{\Theta }}\frac{Bel\left(\theta_{i}\right)+Pl\left(\theta_{i}\right)}{2}\log_{2}\frac{Bel\left(\theta_{i}\right)+Pl\left(\theta_{i}\right)}{2}$ represents the part of measuring discord in the mass function *m*.

Fuzziness is one of the most significant characteristics of fuzzy sets [36]. For the proposed fuzzy belief entropy, we add the interval size to reflect the property of fuzziness. The larger the interval, the stronger the fuzziness of proposition representation, which makes decision-making more difficult. In addition, we also consider the uncertainty in terms of the discord and the nonspecificity. Therefore, the proposed fuzzy belief entropy considers multielement fuzzy subsets and interval size, which can better measure the uncertainty. Moreover, fuzzy belief entropy contains the following properties:

**Property** **1**(probability consistency). *For all $x\in 2^{\Theta }$, if the mass function m obeys probability distribution, then it satisfies that m(x)=Bel(x)=Pl(x) and the cardinal number of each focal element is 1. Thus, we can get that $\frac{Pl\left(x\right)-Bel\left(x\right)}{2}=0$, which means that the value of the fuzziness measure is 0. In this case, the fuzzy belief entropy degenerates to Shannon entropy, and it can be expressed as:*
(17)$$U_{F}\left(m\right)=-\sum_{x\in 2^{\Theta }}m\left(x\right)\log_{2}m\left(x\right)$$

**Property** **2**(non-negativity [52]). *According to the definition of evidence theory in Section 2.2, we can get that 0<Bel(x)≤Pl(x)≤1; then, $0<\frac{Bel(x)+Pl(x)}{2}\leq 1$ and $\frac{Pl\left(x\right)-Bel\left(x\right)}{2}⩾0$. Thus, based on Equation (Equation 15), we can get a conclusion that $U_{F}\left(m\right)⩾0$. In particular, $U_{F}\left(m\right)=0$ if and only if m(x)=1. When the proposition is certain, that is, there is no uncertainty, the value of fuzzy belief entropy is 0.*

## 4. Numerical Example

In the section, some calculation examples are performed and compared with other methods in some particular situations, which verified that the proposed fuzzy belief entropy is effective and persuasive.

### 4.1. Example 1

Given a FOD $\Theta =\left\{x\right\}$, for a fuzzy membership function $m\left(x\right)=1$. Then, it can be converted to
$$Bel\left(x\right)=1,Pl\left(x\right)=1.$$

Thus, the calculation process of Shannon entropy $H_{S}$ and fuzzy belief entropy $U_{F}$ is performed as follows:$$\begin{array}{cc} & H_{S}\left(m\right)=-1\times \log1=0\\ & U_{F}\left(m\right)=-\frac{1+1}{2}\times \left(\log_{2}\frac{1+1}{2\times \left(2^{1}-1\right)}+\frac{1-1}{2}\right)=0 \end{array}$$

### 4.2. Example 2

Given a FOD $\Theta =\left\{x,y,z\right\}$, for a fuzzy membership function $m\left(x\right)\;=\;m\left(y\right)\;=\;m\left(z\right)\;=\;\frac{1}{3}$, then:$$Bel\left(x\right)\;=\;Bel\left(y\right)\;=\;Bel\left(z\right)\;=\;\frac{1}{3},$$
$$Pl\left(x\right)\;=\;Pl\left(y\right)\;=\;Pl\left(z\right)\;=\;\frac{1}{3}.$$

Thus, the calculation process of Shannon entropy $H_{S}$ and fuzzy belief entropy $U_{F}$ can be perfomed as follows:$$\begin{array}{cc} H_{S}\left(m\right) & =\;-\frac{1}{3}\times \log_{2}\frac{1}{3}-\frac{1}{3}\times \log_{2}\frac{1}{3}-\frac{1}{3}\times \log_{2}\frac{1}{3}\;=\;1.5850\\ U_{F}\left(m\right) & =\left(-\frac{\frac{1}{3}+\frac{1}{3}}{2}\times \log_{2}\frac{\frac{1}{3}+\frac{1}{3}}{2\times \left(2^{1}-1\right)}+\frac{\frac{1}{3}-\frac{1}{3}}{2}\right)\\ & +\left(-\frac{\frac{1}{3}+\frac{1}{3}}{2}\times \log_{2}\frac{\frac{1}{3}+\frac{1}{3}}{2\times \left(2^{1}-1\right)}+\frac{\frac{1}{3}-\frac{1}{3}}{2}\right)\\ & +\left(-\frac{\frac{1}{3}+\frac{1}{3}}{2}\times \log_{2}\frac{\frac{1}{3}+\frac{1}{3}}{2\times \left(2^{1}-1\right)}+\frac{\frac{1}{3}-\frac{1}{3}}{2}\right)=1.5850 \end{array}$$

It can be easily observed from example Section 4.1 and example Section 4.2 that, when, when the membership function of IT2 FS is a definite value rather than an interval, that is, the membership function belongs to T1 FS, the uncertainty of the fuzzy sets can also be calculated by Shannon entropy. We clearly observe that the above calculation results of two methods are the same.

### 4.3. Example 3

Given a FOD $\Theta =\;\left\{x,y,z\right\}$, for a fuzzy membership function $m\left(x\right)\;=\;0.5$, $m\left(y,z\right)\in \left[0.3,0.8\right]$, then:$$Bel\left(x\right)\;=\;Pl\left(x\right)\;=\;0.5,$$
$$Bel\left(y,z\right)\;=\;0.3,Pl\left(y,z\right)\;=\;0.8.$$

Thus, the calculation process of Pan & Deng’s new belief entropy $E_{PD}$ [52] and fuzzy belief entropy $U_{F}$ can be performed as follows: $$\begin{array}{cc} & E_{PD}\left(m\right)\;=\;\left(-\frac{0.5+0.5}{2}\times \log_{2}\frac{0.5+0.5}{2\times \left(2^{1}-1\right)}\right)+\left(-\frac{0.3+0.8}{2}\times \log_{2}\frac{0.3+0.8}{2\times \left(2^{2}-1\right)}\right)\;=\;1.8461\\ & U_{F}\left(m\right)=\left(-\frac{0.5+0.5}{2}\times \log_{2}\frac{0.5+0.5}{2\times \left(2^{1}-1\right)}+\frac{0.5-0.5}{2}\right)\\ & \;\;+\left(-\frac{0.3+0.8}{2}\times \log_{2}\frac{0.3+0.8}{2\times \left(2^{2}-1\right)}+\frac{0.8-0.3}{2}\right)=2.0961 \end{array}$$

### 4.4. Example 4

Given a FOD $\Theta =\left\{x,y,z\right\}$, for a fuzzy membership function $m\left(x\right)\;\in \;\left[0.4,0.6\right],\;m\left(y,z\right)\;\in \;\left[0.3,0.8\right]$, then:$$Bel\left(x\right)\;=\;0.4,\;Pl\left(x\right)\;=\;0.6,$$
$$Bel\left(y,z\right)\;=\;0.3\;,Pl\left(y,z\right)\;=\;0.8,$$

Thus, the calculation process of Pan & Deng’s new belief entropy $E_{PD}$ [52] and fuzzy belief entropy $U_{F}$ can be performed as follows: $$\begin{array}{cc} & E_{PD}\left(m\right)\;=\;\left(-\frac{0.4+0.6}{2}\times \log_{2}\frac{0.4+0.6}{2\times \left(2^{1}-1\right)}\right)+\left(-\frac{0.3+0.8}{2}\times \log_{2}\frac{0.3+0.8}{2\times \left(2^{2}-1\right)}\right)\;=\;1.8461\\ & U_{F}\left(m\right)\;=\;\left(-\frac{0.4+0.6}{2}\times \log_{2}\frac{0.4+0.6}{2\times \left(2^{1}-1\right)}+\frac{0.6-0.4}{2}\right)\\ & \;\;+\left(-\frac{0.3+0.8}{2}\times \log_{2}\frac{0.3+0.8}{2\times \left(2^{2}-1\right)}+\frac{0.8-0.3}{2}\right)\;=\;2.1961 \end{array}$$
Comparing the results of Example 3 (Section 4.3) and Example 4 (Section 4.4), we can observe that, although the mean value of interval membership of fuzzy subset *x* is equal to 0.5 for both of them, the interval size of membership degree is larger in Example 3 (Section 4.3), and the uncertainty should be larger. If the uncertainty obtained by using Pan & Deng’s new belief entropy [52], the results are constant, which is obviously unreasonable. It is also demonstrated that the proposed fuzzy belief entropy is more persuasive because it takes the fuzziness caused by the interval into account.

### 4.5. Example 5

Given a FOD Θ, which contains elements which are numbers from 1 to 15, for a fuzzy membership function as follows:$$m\left(3\right)\;\in \;\left[0.15,0.2\right],\;m\left(3,4,5\right)\;\in \;\left[0.1,0.3\right],$$
$$m\left(4,8\right)\;\in \;\left[0.1,0.25\right],\;m\left(A\right)\;\in \;\left[0.7,1\right].$$

In addition, in the experiment, we also used two other methods combined with interval probability to measure the uncertainty of IT2 FS between the regions for comparison.

From the calculations in Table 2, it can be observed that Pan & Deng’s new belief entropy [52] and fuzzy belief entropy increase with the cardinality of subset A increases, and the difference between them is fixed. The difference is determined by the interval size because it adds to the measure of fuzziness. On the other hand, Wang & Song’s UM [61] does not change with the size of fuzzy subset A because the method does not take into account the uncertainty of multivariate subsets [61].

### 4.6. Example 6

As the entropy established based on the belief structure, we also want to test the uncertainty measure of the entropy for the mass function. A frame of discernment, Θ, contains elements which are numbers from 1 to 15. For a fuzzy membership function:$$m\left(3,4,5\right)\;=\;0.05,\;m\left(8\right)\;=\;0.05,\;m\left(A\right)\;=\;0.8,\;m\left(\Theta \right)\;=\;0.1$$

Table 3 displays the change of fuzzy belief entropy with the subset *A* getting larger. It can be observed that fuzzy belief entropy increases with the increase of the element base of subset A, which makes sense.

Furthermore, for the purpose of showing the feasibility of fuzzy belief entropy, the same data are calculated by different methods displayed in Table 3 in this experiment. From the results in Figure 3. It is clear to see that the proposed fuzzy belief entropy, Deng entropy, Wang & Song’s UM, and Dubois & Prade’s UM keep increasing as the cardinal number of subset *A* rises, while the results obtained by other methods will decrease or remain unchanged with the increase of *A*, which is clearly unreasonable. In addition, when the mass function obeys a probability distribution function, Dubois & Prade’s UM cannot reduce to the Shannon entropy [51]. Because Wang & Song’s UM is only related to the measure of discord, it makes the increase of uncertainty small and irregular. Deng entropy cannot directly measure the uncertainty of a given belief interval, so it has certain limitations. Fuzzy belief entropy is essentially improved on the basis of Deng entropy. It can be clearly seen that the Deng entropy and fuzzy belief entropy have the same upward trend. However, because fuzzy belief entropy considers the uncertainty caused by the interval size, the increase of fuzzy belief entropy with the change of subset *A* is larger than that of Deng entropy. The proposed fuzzy belief entropy combines the interval boundary and the cardinal number of each subset of BPA, and measures the uncertainty from the three aspects of fuzziness, discord, and nonspecificity respectively, so the result is more reasonable.

## 5. Conclusions

For IT2 FS, how to measure the uncertainty of IT2 FS is still an open issue. The specific objective of this study to transform the fuzzy membership function into the belief structure, and then propose the fuzzy belief entropy, so as to measure the uncertainty of IT2 FS. Derived from belief structure, it can effectively calculate the uncertainty of multivariate fuzzy subsets. The proposed fuzzy belief entropy is the extension of Shannon entropy. When the membership function belongs to type-1 fuzzy sets, fuzzy belief entropy degenerates to Shannon entropy. Because the uncertainty caused by the size of the interval is considered, fuzzy belief entropy is suitable for calculating the uncertainty of IT2 FS and BPA, which has a wide range of applications. Furthermore, the proposed fuzzy belief entropy measures the uncertainty from the three aspects of fuzziness, discord, and nonspecificity, respectively, so the result is more reasonable. Eventually, the efficiency of fuzzy belief entropy is demonstrated by some examples and comparison with the existing entropy.

## Figures and Tables

**Figure 1 entropy-23-01265-f001:**
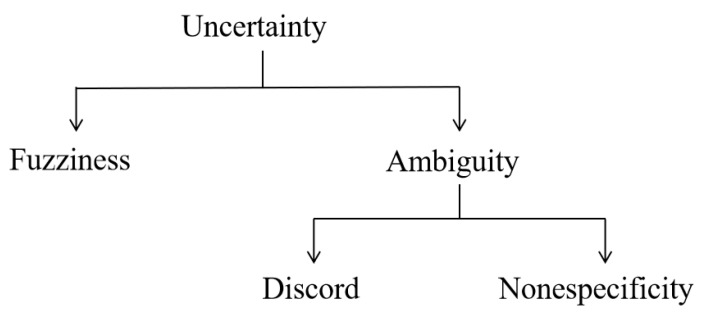
Three basic types of uncertainty. Reprinted with permission from ref. [3]. Copyright 2021 IEEE Xplore.

**Figure 2 entropy-23-01265-f002:**
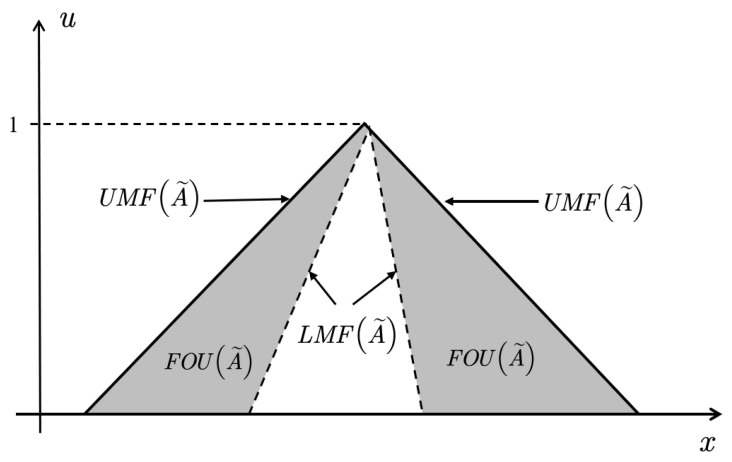
FOU, LMF, UMF for IT2 FS. Reprinted with permission from ref. [22].Copyright 2021 IEEE Xplore.

**Figure 3 entropy-23-01265-f003:**
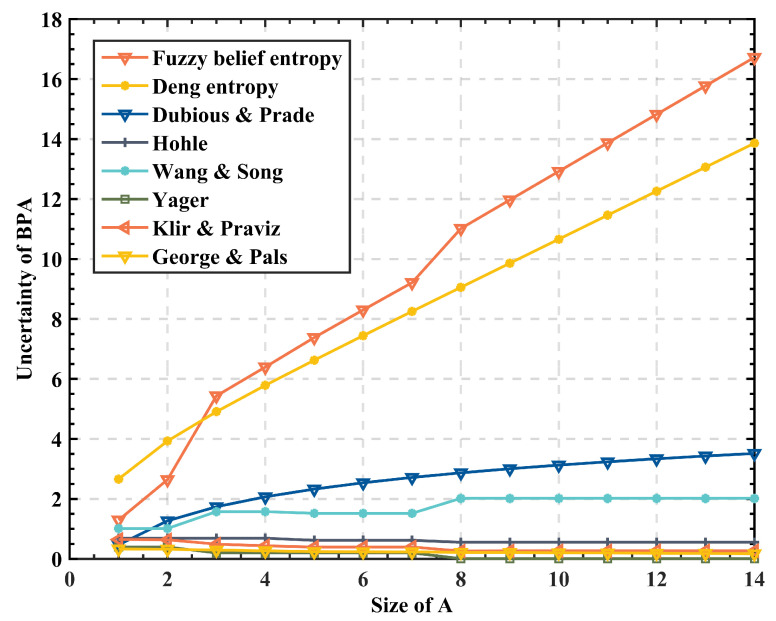
Comparison between fuzzy belief entropy and other methods by using Example 6 (Section 4.6).

**Table 1 entropy-23-01265-t001:** Some existing belief entropy.

Items	Uncertainty Expression
Hole’s UM [56]	$E_{H}\left(m\right)=-\sum_{\theta_{i}\in 2^{\Theta }}m\left(\theta_{i}\right)\log_{2}Bel\left(\theta_{i}\right)$
Yager’s UM [57]	$E_{Y}\left(m\right)=-\sum_{\theta_{i}\in 2^{\Theta }}m\left(\theta_{i}\right)\log_{2}Pl\left(\theta_{i}\right)$
Dubois and Prade’s UM [58]	$E_{DP}\left(m\right)=-\sum_{\theta_{i}\in 2^{\Theta }}m\left(\theta_{i}\right)\log_{2}\left|\theta_{i}\right|$
Klir & Parviz’s strife [59]	$E_{KP}\left(m\right)=-\sum_{A\in 2^{\Theta }}m\left(A\right)\log_{2}\sum_{B\in 2^{\Theta }}m\left(B\right)\frac{\left|A\cap B\right|}{\left|A\right|}$
George & Pal’s conflict measure [60]	$E_{GP}\left(m\right)=-\sum_{A\in 2^{\Theta }}m\left(A\right)\sum_{B\in 2^{\Theta }}m\left(B\right)\left(1-\frac{\left|A\cap B\right|}{\left|A\cup B\right|}\right)$
Wang & Song’s UM [61]	$E_{SU}\left(m\right)=\sum_{A\in 2^{\Theta }}\left[\begin{array}{c} -\frac{Bel\left(A\right)+Pl\left(A\right)}{2}\log_{2}\frac{Bel\left(A\right)+Pl\left(A\right)}{2}\\ +\frac{Pl\left(A\right)-Bel\left(A\right)}{2} \end{array}\right]$
Deng entropy [51]	$E_{D}\left(m\right)=-\sum_{A\in 2^{\Theta }}m\left(A\right)\log_{2}\frac{m\left(A\right)}{2^{\left|A\right|}-1}$
Pan & Deng’s entropy [52]	$E_{PD}\left(m\right)=\sum_{A\in 2^{\Theta }}\left[-\frac{Bel\left(A\right)+Pl\left(A\right)}{2}\log_{2}\frac{Bel\left(A\right)+Pl\left(A\right)}{2\left(2^{\left|A\right|}-1\right)}\right]$

**Table 2 entropy-23-01265-t002:** Comparison among Wang & Song’s method, Pan & Deng’s method and fuzzy belief entropy when *A* changes.

Cases	Wang & Song [61]	Pan & Deng [52]	Fuzzy Belief Entropy
$A=\left\{1\right\}$	1.8787	2.3176	2.7176
$A=\left\{1,2\right\}$	1.8787	3.6648	4.0648
$A=\left\{1,2,3\right\}$	1.8787	4.7038	5.1038
$A=\left\{1,2,3,4\right\}$	1.8787	5.6384	6.0384
$A=\left\{1,2\cdots ,5\right\}$	1.8787	6.5286	6.9286
$A=\left\{1,2\cdots ,6\right\}$	1.8787	7.3983	7.7983
$A=\left\{1,2\cdots ,7\right\}$	1.8787	8.2580	8.6580
$A=\left\{1,2\cdots ,8\right\}$	1.8787	9.1128	9.5128
$A=\left\{1,2\cdots ,9\right\}$	1.8787	9.9652	10.3652
$A=\left\{1,2\cdots ,10\right\}$	1.8787	10.8164	11.2164
$A=\left\{1,2\cdots ,11\right\}$	1.8787	11.6670	12.0670
$A=\left\{1,2\cdots ,12\right\}$	1.8787	12.5173	12.9173
$A=\left\{1,2\cdots ,13\right\}$	1.8787	13.3674	13.7674
$A=\left\{1,2\cdots ,14\right\}$	1.8787	14.2175	14.6175

**Table 3 entropy-23-01265-t003:** Fuzzy belief entropy of the size of A.

Cases	Fuzzy Belief Entropy
$A=\left\{1\right\}$	1.2944
$A=\left\{1,2\right\}$	2.6416
$A=\left\{1,2,3\right\}$	5.4359
$A=\left\{1,2,3,4\right\}$	6.3980
$A=\left\{1,2\cdots ,5\right\}$	7.3814
$A=\left\{1,2\cdots ,6\right\}$	8.3022
$A=\left\{1,2\cdots ,7\right\}$	9.2125
$A=\left\{1,2\cdots ,8\right\}$	11.0186
$A=\left\{1,2\cdots ,9\right\}$	11.9713
$A=\left\{1,2\cdots ,10\right\}$	12.9226
$A=\left\{1,2\cdots ,11\right\}$	13.8733
$A=\left\{1,2\cdots ,12\right\}$	14.8236
$A=\left\{1,2\cdots ,13\right\}$	15.7738
$A=\left\{1,2\cdots ,14\right\}$	16.7239

## Data Availability

Data is contained within the article.

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
