# Peer review of "Uncertainty of Interval Type-2 Fuzzy Sets Based on Fuzzy Belief Entropy"

_entropy, 2021, doi:10.3390/e23101265_

Round 1

Reviewer 1 Report

A novel entropy-based measure applied to evaluate the fuzziness and ambiguity  of IT2 fuzzy sets is proposed.

The authors must highlight more clearly in the introduction what are the weaknesses of the measures of the uncertainty of IT2 FS presented in the literature and, specifically, why the proposed entropic measure is more efficient than them. 

In paragraph 2.1 reference is made to existing belief entropy measures in Tab. 3. Instead, the authors probably intend to refer to the beliefentropy measures in Table 1. It is necessary that the authors deepen the discussion of the existing entropy measures in Tab. 1, highlighting their characteristics, advantages and defects. 

In section 3, the discussion of the proposed belief entropy measure must be further investigated and the importance of its property 1 and property 2 properties must be highlighted.   The comparative results summarized in Fig. 3 need to be explored in depth. The fuzzy belief entropy and the Dang Entropy seem to perform similarly to each other and it is necessary to justify the differences.  

Author Response

[Comment 1] The authors must highlight more clearly in the introduction what are the weaknesses of the measures of the uncertainty of IT2 FS presented in the literature and, specifically, why the proposed entropic measure is more efficient than them. 

Response: Thank you very much for the reminder. In the studies of uncertainty of interval type-2 fuzzy sets, Mendel's method and Green's method have attracted much attention. In the introduction, this paper focuses on the characteristics and shortcomings of the two methods, and points out that they have a common problem: they do not take into account the uncertain influence caused by multiple subsets. The method proposed in this paper based on belief structure can solve this problem well. For details about the modification in the manuscript, see [Pg1-2, Ln33-44 and Ln64-77]

[Comment 2] In paragraph 2.1 reference is made to existing belief entropy measures in Tab. 3. Instead, the authors probably intend to refer to the belief entropy measures in Table 1. It is necessary that the authors deepen the discussion of the existing entropy measures in Tab 1, highlighting their characteristics, advantages and defects. 

Response: Thanks for your kind reminders. We have corrected the reference in section 2.5 to Table1. Fuzzy belief entropy combines the ideas of Wang & Song's UM and Pan & Deng's entropy. We discussed the characteristics and shortcomings of the two methods in Section 2.5[Pg6-7, Ln129-137], and added the following descriptions:

“In addition, Deng entropy considered uncertainty in terms of both Discord and the nonspecificity, but it cannot directly measure the uncertainty for a given belief interval. Wang & Song's UM and Pan & Deng's entropy both use the intermediate value of the belief interval to measure the uncertainty, the former holds that the length of interval can be used as a measure of fuzziness, while the latter is improved on the basis of Deng entropy, so the uncertainty brought by multivariate subsets is considered. But these two aspects are very important to measure the uncertainty of IT2 FS. In this paper, a new entropy is proposed, which measures the uncertainty caused by both interval length and multivariate subset.”

What’s more, in Section 4, we combined the following examples to compare and analyze these existing entropies listed in Table1, and finally demonstrated the feasibility and efficiency of the proposed fuzzy belief entropy.

[Comment 3] In section 3, the discussion of the proposed belief entropy measure must be further investigated and the importance of its property 1 and property 2 properties must be highlighted. The comparative results summarized in Fig. 3 need to be explored in depth. The fuzzy belief entropy and the Dang Entropy seem to perform similarly to each other and it is necessary to justify the differences.  

Response: Thank you very much for the reminder. We made the following adjustments:

First, we described the two properties in detail. The first property mainly tells that when the belief function obeys the probability distribution, the fuzzy belief entropy can be transformed into Shannon entropy, which shows that this method has certain theoretical support. The second property mainly discusses that the value range of fuzzy belief entropy is non-negative, and emphasizes that when the entropy value is 0, it means that there is no uncertainty. You can see the manuscript in [Pg6-7, Ln 155-160].

In addition, we have a more in-depth discussion on Figure 3 in this paper. It is reasonable for fuzzy belief entropy and Deng entropy to show the same upward trend, because fuzzy belief entropy is an improved method based on Deng entropy to solve the uncertainty of belief interval. However, compared with Deng entropy, fuzzy belief entropy takes into account the uncertainty caused by the interval length, so it increases more with the change of subset A. You can see the manuscript in [Pg11, Ln 205-209].

Reviewer 2 Report

The paper discuss a significant issue in fuzzy environment to evaluate the uncertainty.

The paper is well constructed, however, the literature review part is important and must be included in order to illustrate the previous related studies.

Author Response

[Comment 1] The paper is well constructed, however, the literature review part is important and must be included in order to illustrate the previous related studies.

Response: Thank you very much for the reminder. In order to ensure the coherence of the article, we have put the literature review in the introduction. In addition, in this revision, we have described in more detail the previous uncertainty research on the measurement interval type-2 fuzzy sets. You can see the manuscript in [Pg1-2, Ln33-44 and Ln64-73].

Reviewer 3 Report

The paper introduces a new kind of entropy that measures uncertainty in systems where belief mass is defined in the form of fuzzy numbers. The entropy, denoted as fuzzy belief entropy, essentially combines the principles of Pan and Dengs’s entropy and Wang and Song’s entropy. 

Many unusual notations such as “(m)” in equations (10) and (11) - why is the function m in parentheses?

The notation is very inconsistent within the article. For example, the plausibility is denoted as Pl() or pl() even in single expression, see e.g. the last row of Table 1. 

Please clarify how the values Bel(y,z)=0.3 and Pl(y,z) = 0.8 are obtained from known m(x)=0.5 and m(y,z) ∈ [0.3, 0.7] in Example 3. Such clarification should be made also in the following examples.

Please add line numbers to the manuscript if a second round of the review takes place.

“Frame of discriminate” should be “Frame of discernment”.

Please check carefully the formatting of mathematical symbols in text. The symbols should be consistently typeset within the article. For example Probability A should have A typeset in italics everywhere not only in equations.

Some sentences do not sound like correct English: “If probability of A cannot be accurately obtained, but the interval of the probability can be obtained.” Please have a native speaker to edit the manuscript.

Misspelled words:

uncertianty, addtion, studys, attension, Demspter-Shafer, nonspecifificity, realtionship, unerstood, describling, prefromed

Author Response

[Comment 1] Many unusual notations such as “(m)” in equations (10) and (11) - why is the function m in parentheses?

Response: Thank you very much for the reminder. In equations (10) and (11), "(m)" actually does not exist, and we have revised it in the manuscript.

[Comment 2] The notation is very inconsistent within the article. For example, the plausibility is denoted as Pl() or pl() even in single expression, see e.g. the last row of Table 1. 

Response: Thank you very much for the reminder. This is due to our carelessness in writing. We have read through the whole manuscript and expressed the plausibility as "Pl()".

[Comment 3] Please clarify how the values Bel(y,z)=0.3 and Pl(y,z) = 0.8 are obtained from known m(x)=0.5 and m(y,z) ∈ [0.3,0.7] in Example 3. Such clarification should be made also in the following examples.

Response: Thank you very much for the reminder. In Example 3 and 4, “m(y,z) ∈ [0.3,0.8]” is the correct expression . We are very sorry that you couldn’t understand the examples in the article because we made mistakes in writing.

[Comment 4] Please add line numbers to the manuscript if a second round of the review takes place.

Response: Thank you very much for the reminder. we have revised it in the manuscript.

[Comment 5] “Frame of discriminate” should be “Frame of discernment”

Response: Thank you very much for the reminder. We have made revisions accordingly.

[Comment 6] Please check carefully the formatting of mathematical symbols in text. The symbols should be consistently typeset within the article. For example Probability A should have A typeset in italics everywhere not only in equations.

Response: Thank you very much for the reminder. We have made revisions accordingly.

[Comment 7] Some sentences do not sound like correct English: “If probability of A cannot be accurately obtained, but the interval of the probability can be obtained.” Please have a native speaker to edit the manuscript.

Response: Thanks for your kind reminders. We revised the sentence as follows:

“Assume that the probability of A is unknown, but the interval of the probability can be obtained.” [Pg4, Ln104-105]

[Comment 8] Misspelled words: uncertianty, addtion, studys, attension, Demspter-Shafer, nonspecifificity, realtionship, unerstood, describling, prefromed

Response: Thank you very much for the reminder. We have read through the whole manuscript and eliminated all spelling mistakes.

Round 2

Reviewer 1 Report

All my suggestions have been accepted and a broader discussion of the proposed belief entropy measure has been included. I consider this paper publishable in the current form. 

Author Response

[Comment 1] All my suggestions have been accepted and a broader discussion of the proposed belief entropy measure has been included. I consider this paper publishable in the current form. 

Response: Thank you very much for your previous comments that helped us improve this manuscript.

Reviewer 2 Report

The paper has been improved and ready to be published.

Author Response

[Comment 1] The paper has been improved and ready to be published.

Response: Thank you very much for your previous comments that helped us improve this manuscript.

Reviewer 3 Report

The authors resolved all issues pointed out in the first round of review. I found the paper acceptable for publishing after minor review and language corrections.

line 16:  addition -> addition

line 203: “it makes the increase of uncertainty is small and irregular” - leave out “is”

line 205: “Fuzzy belief entropy is essentially is improved on the basis“ - leave out the second “is”

Author Response

[Comment 1] The authors resolved all issues pointed out in the first round of review. I found the paper acceptable for publishing after minor review and language corrections.

line 16:  addition -> addition

line 203: “it makes the increase of uncertainty is small and irregular” - leave out “is”

line 205: “Fuzzy belief entropy is essentially is improved on the basis“ - leave out the second “is”

Response: Thank you very much for your previous comments that helped us improve this manuscript. We have eliminated all spelling mistakes and grammatical mistakes.
